# Hyperbolic Diffusion Functionals on a Ring with Finite Velocity

**DOI:** 10.3390/e27020105

**Published:** 2025-01-22

**Authors:** Marco Nizama

**Affiliations:** Departamento de Fisica, Facultad de Ingenieria and CONICET, Universidad Nacional del Comahue, Neuquen 8300, Argentina; marconizama@gmail.com

**Keywords:** lattice, periodic boundary conditions, Fisher’s information, Shannon’s entropy, Cramér–Rao bound, complexity, power-law

## Abstract

I study a lattice with periodic boundary conditions using a non-local master equation that evolves over time. I investigate different system regimes using classical theories like Fisher information, Shannon entropy, complexity, and the Cramér–Rao bound. To simulate spatial continuity, I employ a large number of sites in the ring and compare the results with continuous spatial systems like the Telegrapher’s equations. The Fisher information revealed a power-law decay of $t^{-\nu }$, with ν=2 for short times and ν=1 for long times, across all jump models. Similar power-law trends were also observed for complexity and the Fisher information related to Shannon entropy over time. Furthermore, I analyze toy models with only two ring sites to understand the behavior of the Fisher information and Shannon entropy. As expected, a ring with a small number of sites quickly converges to a uniform distribution for long times. I also examine the Shannon entropy for short and long times.

## 1. Introduction

The study of conducting media aids in comprehending the propagation of attenuated waves with finite velocity, leading to the development of the Telegrapher’s equations (TEs) [1,2,3,4]. The TEs have diverse applications in scientific fields, such as the generalized Cattaneo–Fick equations, which describe diffusive processes known as hyperbolic diffusion with finite velocity instead of infinite velocity [5,6], line transmission [1,4], neuroscience [7,8], biomedical optics [9], wave penetration in complex conducting media [10,11], and asymptotic diffusion from Boltzmann anisotropic scattering [12,13,14,15]. The TEs have been extended to 2D and 3D for engineering applications [16,17] and have been utilized to model cosmic microwave background radiation with spherically hyperbolic diffusion [18,19]. Studies on finite-velocity diffusion in heterogeneous media have been conducted in [20,21,22,23,24,25]. The TEs are crucial in analyzing damping and the propagation of surface gravity waves on a random bottom [26,27]. Thermal wave propagation has been studied both theoretically and experimentally, with the temperature profile being considered positive. This makes it suitable for analysis using the TEs [28,29].

The references mentioned above focus on the continuous-space TEs. However, exploring discrete systems, such as a one-dimensional lattice, can also be intriguing. Discrete TEs can be characterized by introducing a model with jumps, such as spatial discrete hyperbolic diffusion and next-nearest neighbor hopping, instead of analyzing parabolic diffusion with an unbounded velocity process [2]. In references [30,31], the diffusion lattice problem with periodic boundary conditions was studied traditionally. Another example of the discrete TEs is discussed in [32], where the voltage on a transmission line is examined using a repeated inductor and capacitor system.

In this study, I initially analyze an infinite one-dimensional lattice using different jump models (short- and long-range) represented by various transition matrices T [33,34]. Subsequently, I study a ring with periodic boundary conditions, enabling an analytical solution and eliminating the need for numerical integration on the infinite lattice. A unique aspect of this work is the application of classical information theory to describe the system, including Fisher information, Shannon entropy, and the Cramér–Rao bound [35,36,37,38,39,40,41]. The results reveal intriguing patterns in these measures, demonstrating power-law behavior in the Fisher information and complexity.

The paper is structured as follows: In Section 2, I revisit the non-local master equation in time and its solution on a one-dimensional lattice with periodic boundary conditions. Section 3 discusses various jump models in the master equation, including the next-nearest neighbor model, the geometric jump model, and the Poisson jump model. Section 4 defines the Fisher information for discrete systems, as well as the complexity measure and the Cramér–Rao bound. Section 5 presents numerical results for the classical theory measures defined in the previous section. Finally, Section 6 provides conclusions and perspectives on the model. Additionally, the appendices include toy models with two sites in a ring to illustrate Fisher behavior in both the ballistic and diffusive regimes. I also analyze the entropy behavior for all the models studied.

## 2. Evolution Formalism in a Lattice

This section explores the formalism of evolution on a lattice (or ring), starting with a one-dimensional lattice.

### 2.1. Non-Local Master Equation with Finite Velocity

In this section, we examine a master equation for non-local discrete time and spatial processes (where the Kernel Φ(t) is an exponential function) [2,33]:(1)$$\partial_{t}P_{s}\left(t\right)=\int_{0}^{t}\Phi \left(t-t^{\prime }\right)\sum_{s^{\prime }\epsilon D_{L}}H_{ss^{\prime }}P_{s^{\prime }}\left(t^{\prime }\right)dt^{\prime }=\alpha^{2}\int_{0}^{t}e^{-(t-t^{\prime })/\tau }\sum_{s^{\prime }\epsilon D_{L}}H_{ss^{\prime }}P_{s^{\prime }}\left(t^{\prime }\right)dt^{\prime }.$$

The conditional probability is denoted by $P_{s}\left(t\right)=P_{s}\left(t|s_{0},t_{0}\right)$ with $t_{0}=0$, and $s_{0}\in D_{L}$ (representing the lattice domain). Equation (Equation 1) corresponds to the continuous-time random walk (CTWR) model [33]. The condition $\partial_{t}P_{s}{\left(t\right)|}_{t=0}=0$ is implicit, making the conditional probability symmetric with respect to *s*.

The matrix H is connected to the Markov matrix T through the relationship H=T−I. For further information, see [34].

Equation (Equation 1) is linked to classical probability, where the kernel Φ(t) must satisfy the following condition (this condition was established in [34] using the renewal theory to guarantee the positivity of the waiting time function):(2)2ατ<1.

Otherwise, the solution of (Equation 1) may become negative. For example, some values of $P_{s}\left(t\right)$ may be negative. There are parameter regions that do not fulfill this condition, where the probability distribution (meaning $P_{s}\left(t\right)\geq 0$) may exhibit wave-like behavior [34]. In Appendix A of Reference [34], the physical interpretation of the condition (Equation 2) is elucidated using renewal theory following Equation (A7) in Reference [34].

The non-local solution can be expressed as a second derivative in time:(3)$$\partial_{t}^{2}P_{s}\left(t\right)+\frac{1}{\tau }\partial_{t}P_{s}\left(t\right)=\alpha^{2}\sum_{s^{\prime }\in D_{L}}H_{ss^{\prime }}P_{s^{\prime }}\left(t^{\prime }\right).$$

This equation accounts for both wave and diffusive behavior on the discrete lattice. It represents the discrete Telegrapher’s equation, where the term $\left(\frac{1}{\tau }\right)$ relates to energy absorption and $\alpha^{2}$ to the speed squared.

### 2.2. Solution of Non-Local Master Equation

To obtain the solution of (Equation 3), we can use the Fourier and Laplace transformations:(4)$$P_{k}\left(u\right)=\sum_{s=-\infty }^{\infty }e^{iks}\int_{0}^{\infty }e^{-ut}P_{s}\left(t\right)dt.$$

Replacing the last equation in (Equation 3), we obtain the following:(5)$$P_{k}\left(u\right)=\frac{\left(u+\tau^{-1}\right)P_{k}\left(t=0\right)}{u\left(u+\tau^{-1}\right)+\alpha^{2}\left(1-T\left(k\right)\right)},$$
given the initial conditions$$\begin{array}{ccc} P_{s}{\left(t\right)|}_{t=0}=\delta_{s,s_{0}}, & \mathrm{and} & \partial_{t}P_{s}{\left(t\right)|}_{t=0}=0. \end{array}$$
where $P_{k}\left(t=0\right)=e^{iks_{0}}$ and $s_{0}$ is the position of the localized initial condition. The term *T*(*k*) represents the Fourier transforms of elements $T_{s,s^{\prime }}$, the procedure of which can be found in Ref. [34].

The inverse Laplace is given by the following (for an initial condition localized in the site $s_{0}$, we have $P_{k}\left(t=0\right)=e^{iks_{0}}$):(6)$$P_{k}\left(t\right)=\left[\frac{1}{\tau }\left(\frac{e^{iu_{+}}-e^{iu_{-}}}{u_{+}-u_{-}}\right)+\left(\frac{u_{+}e^{iu_{+}}-u_{-}e^{iu_{-}}}{u_{+}-u_{-}}\right)\right]P_{k}\left(t=0\right),$$
where $u_{\pm }$ is pole of denominator of Equation (Equation 5), given by the expression(7)$$u_{\pm }=\frac{1}{2\tau }\left[-1\pm \sqrt{1-{\left(2\tau \alpha \right)}^{2}\left(1-T\left(k\right)\right)}\right].$$

The solution $P_{s}\left(t\right)$ strongly depends on the jump model that has been analyzed.

### 2.3. Ring’s Solution

Finding the solution $P_{s}\left(t\right)$ in space and time is a challenging task due to its dependence on the structure of T(k). We analyze three jump models in a ring, which has an analytical solution using translation invariance. This means that $P_{s+N}\left(t|s_{0},t_{0}\right)=P_{s}\left(t|s_{0},t_{0}\right)$, where *N* is the number of sites in the finite ring. The solution can be expressed as follows:(8)$$P_{s}\left(t\right)\equiv P_{s}\left(t|s_{0},t_{0}\right)=\frac{1}{N}\sum_{r=1}^{N}e^{-i\frac{2\pi r}{N}s}P^{0}\left(k=\frac{2\pi r}{N},t-t_{0}\right),$$
where $P^{0}\left(k,t-t_{0}\right)$ is defined by (Equation 6). Subsequently, we obtain the following [34]:(9)$$P_{s}\left(t\right)=\frac{1}{N}\sum_{r=1}^{N}e^{-i\frac{2\pi r}{N}\left(s-s_{0}\right)}{\left[\frac{1}{\tau }\left(\frac{e^{iu_{+}}-e^{iu_{-}}}{u_{+}-u_{-}}\right)+\left(\frac{u_{+}e^{iu_{+}}-u_{-}e^{iu_{-}}}{u_{+}-u_{-}}\right)\right]}_{k=2\pi r/N},$$
where $u_{\pm }$ is determined by (Equation 7) and is influenced by the jump model on the ring through T(k).

## 3. Jump Models Used on the Finite Ring

In this section, I explore three distinct models of jumps in the ring, encompassing both short and long jump variations. For more information about these jump models, refer to [33,34].

### 3.1. The Next-Nearest Neighbor Jump Model

The referential case is the model with jumps only to first neighbors. The matrix T is expressed simply as follows:(10)$$T=(E^{+}+E^{-})/2.$$

Here, $E^{\pm }$ represents the translation operator for the right (+) and left (−) directions. Applying $E^{\pm }$ to a vector $f_{s}$ results in the following:(11)$$E^{+}f_{s}=f_{s\pm 1}.$$

The Fourier transform of elements $T_{s,s^{\prime }}$ in a one-dimensional lattice is T(k)=cos(k). Substituting this into (Equation 7), we obtain(12)$$u_{\pm }=\frac{1}{2\tau }\left[-1\pm \sqrt{1-{\left(2\tau \alpha \right)}^{2}\left(1-\cos\left(k\right)\right)}\right].$$

To calculate $P_{s}\left(t\right)$ in the ring, we need to substitute the last equation into (Equation 9).

### 3.2. The Geometric Jump Model

In this model, the jumps are not limited to just the immediate neighbors, but also include longer jumps. This is represented by a transition matrix T given by the following:(13)$$T_{s,s^{\prime }}=\frac{1-\gamma }{2\gamma }\left(\gamma^{|s-s^{\prime }|}-\delta_{s,s^{\prime }}\right),$$
where 0<γ<1. By taking the Fourier transform of the above equation, we obtain(14)$$T\left(k\right)=\frac{1-\gamma }{\gamma }\left(\frac{1-\gamma \cos(k)}{1-2\gamma \cos\left(k\right)+\gamma^{2}}-1\right).$$

Following a similar procedure as in the previous subsection, we can derive $P_{s}\left(t\right)$ for the ring by using T(k) in (Equation 7).

### 3.3. The Poisson Jump Model

The final model examined is the long-range jump, with a transition matrix given by(15)$$T_{s,s^{\prime }}=\frac{e^{-\theta }}{2(1-e^{-\theta })}\left(\frac{\theta^{|s-s^{\prime }|}}{|s-s^{\prime }|!}-\delta_{s,s^{\prime }}\right),$$
where θ>0. The Fourier transform is then obtained as follows:(16)$$T\left(k\right)=\frac{e^{-\theta }}{2(1-e^{-\theta })}\left(\exp\left(\theta e^{ik}\right)+\exp\left(\theta e^{-ik}\right)-2\right).$$

The same procedure as in the previous subsection is followed to obtain $P_{s}\left(t\right)$.

## 4. Classical Information Theory: Shannon Entropy and Fisher Information

To analyze the various stages in the time evolution of probability within the ring system, we compute the Shannon entropy. This entropy quantifies the level of disorder present in the system and can be expressed for a discrete system as follows [33,35]:(17)$$S\left(t\right)=-\sum_{s=1}^{N}P_{s}\left(t\right)\ln\left[P_{s}\left(t\right)\right].$$

In this work, we calculate $P_{s}\left(t\right)$ using Formula (Equation 9) for various jump models on a small ring, as previously carried out in [34]. We extend this calculation to multiple sites in the ring in our current study.

Another way to characterize the systems under study is through the Fisher information *I* [36,37,38,39]. This measure quantifies the level of disorder in the systems, where strong disorder indicates a lack of predictability in the spatial variable across its range. A narrow probability distribution corresponds to a high value of *I*, while a broad probability distribution corresponds to a low value of *I*.

In the case of a discrete spatial variable, the Fisher information can be defined as follows (for the ring with sites s=1,⋯,N, we consider $P_{N+1}\left(t\right)=P_{1}\left(t\right)$ and the derivative with respect to *s* as $P_{s+1}\left(t\right)-P_{s}\left(t\right)$, analogous to the continuous spatial case):(18)$$\begin{array}{ccc} I(t) & = & \sum_{s=1}^{N}\frac{{\left[\left(E^{+}-I\right)P_{s}\left(t\right)\right]}^{2}}{P_{s}\left(t\right)}\\ & = & \sum_{s=1}^{N}\frac{{\left[P_{s+1}\left(t\right)-P_{s}\left(t\right)\right]}^{2}}{P_{s}\left(t\right)}\\ & = & \sum_{s=1}^{N}\frac{P_{s+1}^{2}\left(t\right)-2P_{s+1}\left(t\right)P_{s}\left(t\right)+P_{s}^{2}\left(t\right)}{P_{s}\left(t\right)}\\ & = & \sum_{s=1}^{N}\frac{P_{s+1}^{2}\left(t\right)}{P_{s}\left(t\right)}-1. \end{array}$$

In the previous expression, we utilized translational invariance in the finite ring and normalization condition for $P_{s}\left(t\right)$.

We introduce the complexity measure C(t), which combines the Shannon entropy and Fisher information [37,38]:(19)$$C\left(t\right)=\frac{e^{2S(t)}I\left(t\right)}{2\pi e}\geq 1.$$

It is a known fact that the complexity equals one for a diffusive process. For more information, please see Appendix C or reference [41].

The Cramér–Rao bound (or inequality) is defined as follows [40,42]:(20)$$CR\left(t\right)=I\left(t\right)\Delta x{\left(t\right)}^{2}\geq 1.$$

The dispersion is denoted by $\langle x{\left(t\right)}^{2}\rangle =\Delta x{\left(t\right)}^{2}$. The equation for the dispersion is given by the following:(21)$$\Delta x{\left(t\right)}^{2}=\sum_{s=1}^{\infty }{\left(s-s_{0}\right)}^{2}P_{s}\left(t\right).$$

The Cramér–Rao bound is a crucial tool in optimal experimental design. The connection between estimator variance and Fisher information shows that minimum variance corresponds to maximum Fisher information. For a diffusive process, the Cramér–Rao bound reaches its minimum value of one. In Ref. [34], the expression for $\Delta x{\left(t\right)}^{2}$ is derived analytically for all the jump models considered.

In this study, I focus on calculating measures from classical information theory. We will consider the case where the total number of sites in the ring, *N*, is much larger than the 10 sites (as considered in Ref. [34]) to analyze long-time values and avoid the scenario where $P_{s}\left(t\right)$ approaches 1/N as *t* approaches infinity. This ensures that in Equation (Equation 18), we do not reach $I\left(t\to \infty \right)=[N{\left(1/N\right)}^{2}/\left(1/N\right)-1]\to 0$, while also using translational invariance, representing the system with the highest disorder.

## 5. Numerical Results for Functionals in a Ring

In this section, I present the main numerical results for the Fisher information, complexity, Shannon entropy, and the Cramér–Rao bound for various jump models discussed in Section 3 for a finite ring of size *N* = 401 sites.

In Figure 1, we display the Fisher information *I* as a function of time (in arbitrary units) using (Equation 18) for three models of jumps in the ring: the next-nearest neighbor jump model (NN model), the geometric jump model (GJ model), and the Poisson jump model (PJ model) discussed in Section 3. In the ballistic regime where time values are small (i.e., t≪τ), the Fisher information has been fitted with a power-law function $I=at^{-2}$. It is noted that this is proportional to $1/\Delta x{\left(t\right)}^{2}$ (as defined by (Equation 21)), which is a novel finding in this study.

For the NN model, the fitted parameter is a=4. For the GJ model, we obtain *a* = 7.8, and for the PJ model, the result is a=5. However, for longer times (t≫τ, in the diffusive regime), *I* is fitted with a different exponent $I=D^{-1}t^{-1}$, where *D* is the dissipative parameter—i.e., $I=1/\Delta x{\left(t\right)}^{2}$ in the diffusive regime (see Appendix B). This result is well known for the spatial-continuous TEs, as shown in Ref. [41]. The dissipative parameters were obtained in reference [34]. For the NN model, $D=\alpha^{2}\tau$; for the GJ model, $D=\frac{\alpha^{2}\tau \left(1+\gamma \right)}{{\left(1-\gamma \right)}^{2}}$; and for the PJ model, $D=\frac{\alpha^{2}\tau \theta \;e^{\theta }\left(1+\theta \right)}{\left(e^{\theta }-1\right)}$. It is important to note that the the dissipative parameter value varies depending on the jump model used. The parameters used in the models are as follows: τ=0.4, α=1, γ=0.4, and θ=0.4.

Another important finding was determining the transition from the ballistic regime to the diffusive regime by finding the crossover point where the fitted functions $I_{\mathrm{ballistic}}=at^{-2}$ and $I_{\mathrm{diffusive}}=1/D^{-1}t^{-1}$ intersect. In the NN model, the crossover time $t_{c}$ is approximately 1.6, in the GJ model, it is around 11, and in the PJ model, it is roughly 3.3. This analysis aids in characterizing the shift from the ballistic to diffusive regime, even though the crossover is gradual.

In essence, the Fisher information as a function of time can be described by power-law functions for small and large time values compared with τ, indicating a consistent proportionality to $1/\Delta x{\left(t\right)}^{2}$ across all models. This implies a consistent pattern in Fisher behavior across different jump models in the ring. However, there is a deviation from power-law behavior for intermediate time values. Additionally, the transition time $t_{c}$ from the ballistic to diffusive regimes has also been established for all studied jump models.

In Figure 2, the complexity $C\left(t\right)$ is shown as a function of time for a ring with periodic boundary conditions having N=401 sites. The solid lines in black, red, and blue represent the NN, GJ, and PJ jump models, respectively. We performed a fitting analysis of the complexity as a function of time and obtained the following function for the three jump models: $C_{fitted}\left(t\right)=1+at^{-b}$ for all t>0. The values of *a* are 0.288 for the NN model, 0.495 for the GJ model, and 0.353 for the PJ model. The value of *b* is approximately 1.95 for all three jump models.

In the regime where *t* is much smaller than τ, the complexity behavior follows a power-law of $t^{-b}$ (ballistic regime) with b approximately equal to two, a novel result for the ring system. Conversely, for t≫τ, the complexity converges to a value of one (diffusive regime), a well-known result for the continuous TEs. Additional details can be found in Appendix C and Ref. [41]. The parameters used are as follows: τ=0.4 and α=1. γ=0.4 corresponds to the GJ model, and θ=0.4 corresponds to the PJ model. The fitted function $C_{fitted}\left(t\right)$ does not accurately represent the behavior of C(t) for intermediate time values.

In summary, the complexity shows different behaviors depending on τ: for short times, it follows a power-law function, indicating ballistic behavior, while for long times, it remains constant at one, indicating diffusive behavior. These patterns were observed in all three models analyzed.

In Figure 3, we show the Fisher information as a function of Shannon entropy for three jump models: the NN model in black, the GJ model in red, and the PJ model in blue solid lines. For small values of *S* (equivalent to short times, i.e., t≪τ), the function *I* can be fitted by a power-law function $I\left(S\right)\approx aS^{-1.12}$. This discovery provides a new insight into the ring system. The values of *a* for the NN, GJ, and PJ models are 9.25, 18.67, and 12.73, respectively. This relationship can be expressed as the product of I and a power-law of S (similar to Refs. [37,38]):(22)$$IS^{1+0.12}=constant.$$
The constant value depends on the jump models being studied. This relationship between *I* and *S* can be explained using a simple toy model, such as a ring with only two sites (refer to Appendix A), where I.S≈constant. The correction in the exponent of S can be made by considering more than two sites in the ring.

By using (Equation 22) and the relation $I\propto t^{-2}$ as observed in Figure 1, for t≪τ, we can derive an expression for the entropy in the form of a power law. This result is consistent with the *S* vs. *t* plot shown in Appendix C. The expression is as follows:(23)$$S\propto t^{1.67}.$$

For large values of *S* (corresponding to long *t* values, i.e., t≫τ), as shown in Figure 3 represented by green lines, the Fisher behavior of the three models studied fits an exponential function, indicating the diffusive regime. This is evident in the complexity formula for the diffusive regimen, which is $e^{2S}I/\left(2\pi e\right)=1$. This can be expressed as $I=\left(2\pi e\right)e^{-2S}$ (for more details, see Appendix C) [41]. The parameters used are τ=0.4 and α=1, with the GJ model corresponding to γ=0.4 and the PJ model corresponding to θ=0.4.

In summary, high Fisher information, indicating low entropy (less disorder), is associated with a power-law relationship with entropy, which is a new finding for the ring. Conversely, low Fisher information, signifying high entropy (more disorder), is linked to an exponential relationship with entropy.

In Figure 4, we present the Cramér–Rao bound (Equation 20) over time for the three models of studied jumps: the NN model, the GJ model, and the PJ model are depicted by black, red, and blue solid lines, respectively. When t≪τ (ballistic regimen), the Fisher information scales as $t^{-2}$ (refer to Figure 1), while $\Delta x{\left(t\right)}^{2}\approx \alpha^{2}t^{2}/2$, leading to a constant value of CR greater than 1. This novel finding for the ring indicates a high Fisher information value, suggesting the ballistic regimen. Conversely, for t≫τ (diffusive regime), the Cramér–Rao bound converges to the lower bound of one (diffusive regime) [41]. The parameters used are τ=0.4 and α=1. When γ=0.4, it corresponds to the GJ model, and for θ=0.4, it corresponds to the PJ model.

In summary, the Cramér–Rao bound is greater than one (with a constant value) in the ballistic regime and approaches one in the diffusive regime. This measure exhibits a consistent trend over time.

## 6. Conclusions

This study focuses on discrete non-local master equations. Instead of calculating the probability distribution $P_{s}\left(t\right)$ for a lattice, a ring with 401 sites was used for simplicity. Two types of jump models were examined: short- and long-range transition matrices. The solution for $P_{s}\left(t\right)$ in the ring was obtained through discrete Fourier transform, enabling an analytical representation.

In the study of different jump models, the Fisher functional I(t) and Shannon functional S(t) were employed to characterize the discrete spatial system in the generalized discrete TEs. The solution of the generalized discrete TEs shows ballistic behavior for all jump models at short times (t≪τ) and diffusive behavior at long times (t≫τ), similar to the continuous spatial limit. As time approaches infinity (t→∞), $P_{s}\left(t\right)$ converges to 1/N (uniform distribution function), with *I* approaching zero.

The Fisher functional captures these behaviors by showing a $t^{-2}$ proportionality for t≪τ and a $D^{-1}t^{-1}$ fit for t≫τ. Here, *D* represents the dissipative parameter. In the NN jump model, $D=\tau \alpha^{2}$, in the GJ model, $D=\frac{\alpha^{2}\tau \left(1+\gamma \right)}{{\left(1-\gamma \right)}^{2}}$, and in the PJ model, $D=\frac{\alpha^{2}\tau \theta \;e^{\theta }\left(1+\theta \right)}{\left(e^{\theta }-1\right)}$ [34]. As *t* approaches infinity, the Fisher functional tends to zero due to the uniform distribution of $P_{s}$.

The complexity as a function of time confirms these observed regimes, with the fitting power-law function for small values of time (t≪τ, in the ballistic regime). Conversely, for long values of time (t≫τ), it behaves differently (non-power-law), tending to a constant value of one (diffusive regime). We have also analyzed Fisher information as a function of *S*, showing a power-law behavior for short times, but a non-power-law for long times, exhibiting an exponential behavior connected with the complexity.

The Cramér–Rao bound is a measure used to characterize the systems under study. It can be thought of as an uncertainty principle in classical statistics. It has a constant value greater than one for t≪τ and tends to one for t≫τ.

Additional information can be found in the appendices, where we investigate the Fisher information for the ballistic and diffusive regimes using two toy models (a ring with two sites). We also examine the Shannon entropy over time to distinguish between the two regimes: the ballistic regime follows a power-law, while the diffusive regime follows a logarithmic function. This clarifies why the complexity is equal to one in the diffusive regime.

More research is required to study cases with different initial conditions across the ring, such as random initial conditions, to assess their impact on dynamics in comparison to localized initial conditions. If the condition 2ατ<1 is not met, $P_{s}\left(t\right)$ could become negative. However, for certain values of τ and α, and with specific T(k) structures, $P_{s}\left(t\right)$ may remain non-negative, allowing for the application of classical information theory. This opens up another area for further investigation, which can be analyzed using Bochner’s theorem [33].

## Figures and Tables

**Figure 1 entropy-27-00105-f001:**
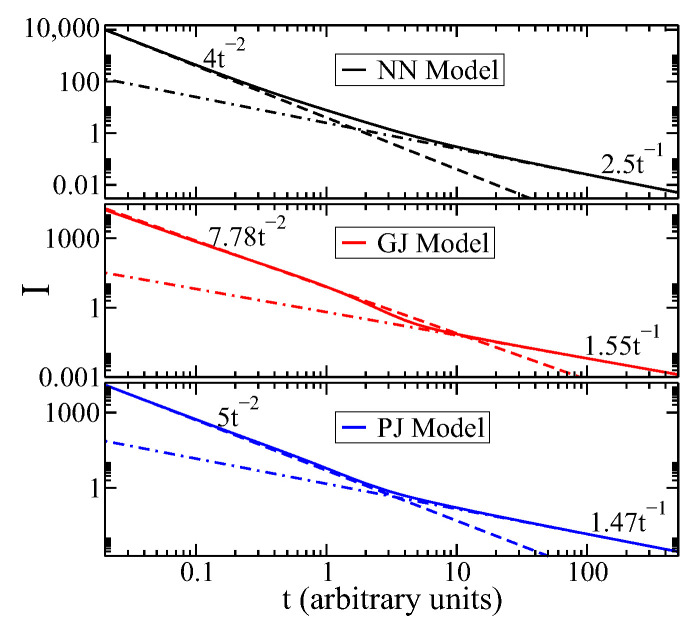
The Fisher information over time in arbitrary units is depicted for N=401 sites in (Equation 9) of the ring. The solid lines represent the NN, GJ, and PJ jump models (black, red, and blue, respectively). In all cases, the dashed lines show a power-law fit of $t^{-2}$ for t≪τ (ballistic regime), while for t≫τ, the Fisher measure fits as $D^{-1}t^{-1}$ in dotted and dashed lines (diffusive regime). The parameters used are τ=0.4, α=1, γ=0.4 (GJ model), and θ=0.4 (PJ model). The plots display the fitted functions, such as $\beta t^{-\nu }$, where ν=2 in the ballistic regime and ν=1 in the diffusive regime, along with the jump models that were studied.

**Figure 2 entropy-27-00105-f002:**
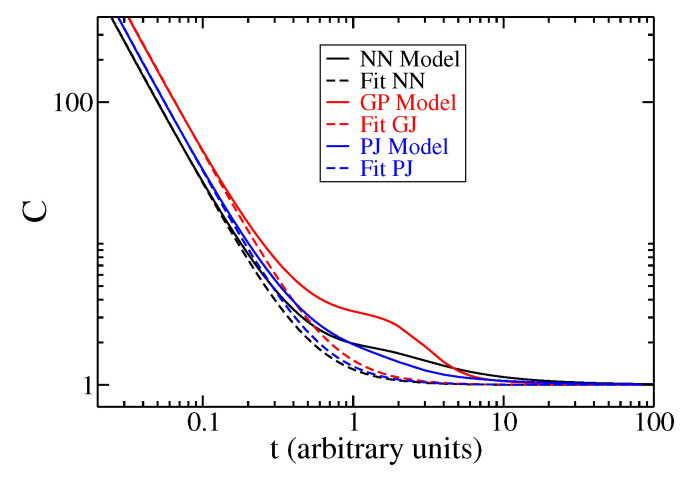
Complexity C(t) as a function of time for three different jump models: NN, GJ, and PJ (represented by solid black, red, and blue lines, respectively). The dashed lines depict the power-law fit $C_{\mathrm{fitted}}\left(t\right)=1+at^{-b}$ for all models, the manuscript providing the values of *a* and *b*. The parameters used are the same as in Figure 1. For small *t* compared to τ, C(t) scales as $t^{-b}$ with *b* close to 2, indicating a ballistic regime. As *t* becomes larger than τ, C(t) approaches a value of 1, signifying the diffusive regime.

**Figure 3 entropy-27-00105-f003:**
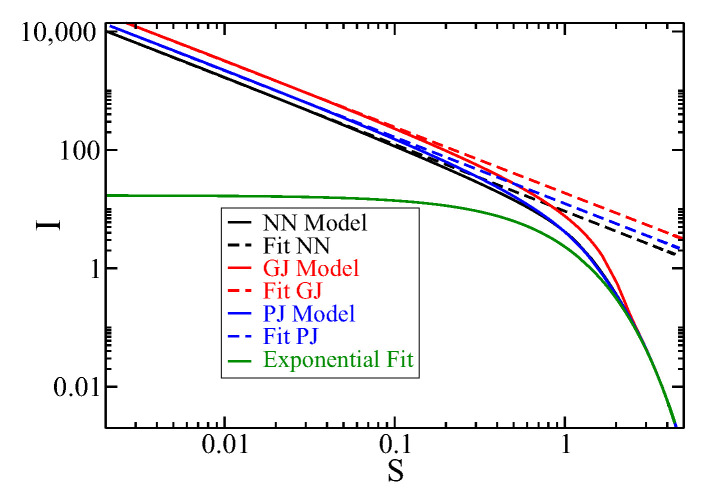
Fisher information is shown plotted against Shannon entropy for the NN model (black line), GJ model (red line), and PJ model (blue line). The dashed lines represent a power-law fit $I=aS^{-1.12}$ for small values of *S*, where a=9.25 (NN model), a=18.67 (GJ model), and a=12.73 (PJ model). For large values of *S*, the Fisher information for all models follows a diffusive regime given by $I=\left(2\pi e\right)e^{-2S}$, shown in green lines. The parameters used are consistent with those in Figure 1.

**Figure 4 entropy-27-00105-f004:**
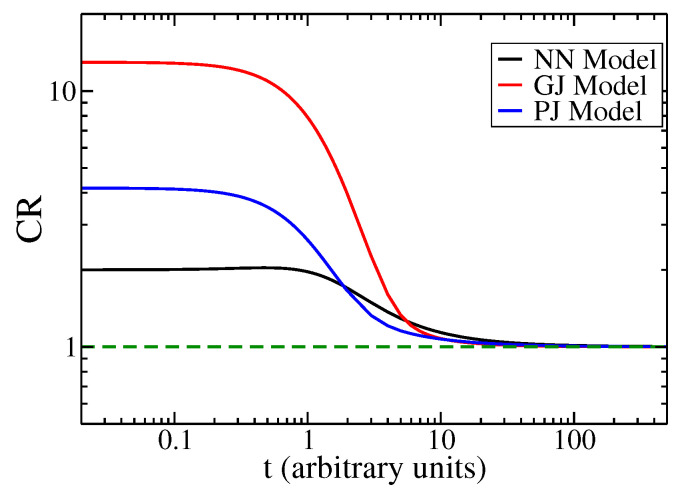
The Cramér–Rao bound (CR) is plotted as a function of *t* for the NN model (black line), GJ model (red line), and PJ model (blue line). The green dashed line indicates the CR value in the diffusive regime. The parameters are the same as those in Figure 1. CR exceeds one for t≪τ (ballistic regime) and approaches one for t≫τ (diffusive regime).

## Data Availability

The data that support the findings of this study are available from the corresponding author upon reasonable request.

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
