# Peer review of "Hyperbolic Diffusion Functionals on a Ring with Finite Velocity"

_entropy, 2025, doi:10.3390/e27020105_

Round 1
Reviewer 1 Report
Comments and Suggestions for Authors
The Author considers an interesting problem of telegrapher’s model with jumps. The topic is worth investigating and of interest to the journal readership. However, the presentation in the manuscript should be improved.
Here are some comments:
- The term “Hyperbolic Diffusion” is included in the title of the manuscript, while only once this term is mentioned in the manuscript (in the introduction), so this should be clarified and discussed in the manuscript.
- TE in complex/heterogeneous media have been also considered in other works, see for example [Physical Review E 93, 052107 (2016)], [The European Physical Journal B 90, 1 (2017)], [Entropy 26, 665 (2024)], [Phys. Rev. E 100, 052147 (2019).], and other related papers.
- Could you please elaborate more what does the condition (2) physically mean?
- Since \alpha corresponds to the speed v, and \tau is the time parameter, the Author could mention that v^2*\tau=D is the diffusion coefficient.
- Section 3: Have these jump models been investigated in diffusion processes previously? If yes, could you refer to the corresponding references and discuss the results? More details for these jump models are needed in order to be clear for most of the readers.
- What are the corresponding mean squared displacements for such jump processes? Can one obtain some analytical results from the PDF given in Section 2.2?
- The presentation in Section 5 should be improved. This part is very important but difficult to follow. The new results there should be stressed and comparison with some known results for the limiting cases should be mention/discussed.
- Some important results are given in appendices. I think that they should be incorporated in the main text.
Author Response
Comments 1: “The term “Hyperbolic Diffusion” is included in the title of the manuscript, while only once this term is mentioned in the manuscript (in the introduction), so this should be clarified and discussed in the manuscript.”
Response 1: Thank you for pointing this out. I have included this term to offer additional clarification.
Comments 2: “TE in complex/heterogeneous media have been also considered in other works, see for example [Physical Review E 93, 052107 (2016)], [The European Physical Journal B 90, 1 (2017)], [Entropy 26, 665 (2024)], [Phys. Rev. E 100, 052147 (2019).], and other related papers.”
Response 2: Thank you for your suggestion. The references [20] and [21] have been included in the manuscript.
Comments 3: “Could you please elaborate more what does the condition (2) physically mean?”
Response 3: Thank you for pointing this out. The physical interpretation of condition (2) is related to renewal theory, as discussed in the recent reference [34] (Phys. Rev. E 110, 024141 (2024)). In a like-diffusion process, the flight time 1/α represents the time for a walker on the lattice to move from one site to another, while the mean waiting time <t> = 1/(α^2τ) indicates the time spent by the walker at a single site. When (1/α) << 1/(α^2τ), condition (2) is met, showing that the walker spends more time at one site than hopping to another. I have referenced where readers can find a more detailed explanation in the manuscript, as I have chosen not to delve into the complexities of renewal theory in this paper.
Comments 4:”Since \alpha corresponds to the speed v, and \tau is the time parameter, the Author could mention that v^2*\tau=D is the diffusion coefficient.”
Response 4: Thank you for your suggestion. I have added a clarification in the manuscript to indicate where the diffusion coefficient was calculated for all the models studied. I have also included a new phrase after the introduction of Equation (21) to address this issue.
Comments 5: ”Have these jump models been investigated in diffusion processes previously? If yes, could you refer to the corresponding references and discuss the results? More details for these jump models are needed in order to be clear for most of the readers.”
Response 5: Thank you for your question. Yes, the jump models have been studied in the new reference [34], which is included at the beginning of section III.
Comments 6: What are the corresponding mean squared displacements for such jump processes? Can one obtain some analytical results from the PDF given in Section 2.2?
Response 6: Thank you for your question. I have added a clarification phrase to specify the reference in which the squared displacements were calculated in analytical form, following the new Equation (21).
Comments 7:”The presentation in Section 5 should be improved. This part is very important but difficult to follow”
Response 7: Thank you for pointing this out. Section 5 has been revised to enhance clarity for our readers.
Comments 8: ”Some important results are given in appendices. I think that they should be incorporated in the main text.”
Response 8: Thank you for your suggestion, but I believe it is not necessary to move the appendices to the main text as it would make the main text longer.
Reviewer 2 Report
Comments and Suggestions for Authors
Report on the manuscript entropy-3385847
"Hyperbolic Diffusion Functionals on a Ring with Finite Velocity"
by Marco Nizama
This paper presents an investigation about a motion on a lattice with periodic boundary conditions governed by a non-local time-evolving master equation. Different system regimes are considered, by using classical theories as Fisher’s information, Shannon’s entropy, complexity, and the Cramér-Rao bound. A toy model with only two ring sites is studied in details.
The subject of the article is suitable for publication on the journal Entropy (MDPI). The conclusions are sound and justified by the data. The manuscript presents some novel results. The title reflects the content of the paper clearly and sufficiently. The presentation, organization and length are satisfactory. The paper is interesting and demonstrates potential to influence applications. The quality of the English is satisfactory. The abstract and summary are properly informative. The illustrations and tables are necessary and acceptable. The references are adequate and all necessary.
However, the following improvements are required:
- The keywords are missing: three to ten pertinent keywords specific to the article should be included.
- Page 2, l. 74-75, Please, improve/clarify the two sentences, since they seem to be in disagreement.
- Page 2, eq. (4): if this is the definition of the double Fourier-Laplace transform of P_s(t) then it should be renamed by using a different symbol, for instance \tilde P instead of P in the left-hand-side, in order to avoid confusion with the expression of the conditional probability. Moreover, there is misprint: the argument of the function is (u) rather than (t).
- Page 2, l. 83: I suggest to display the two initial conditions in the center of the line in order to give them more visibility.
- Page 3, l. 87: The inverse the Laplace -> The inverse of the Fourier-Laplace transform
- Page 5, the equations (18) and (19) are not recalled in the paper, so that the numbers can be cancelled
- Page 5, equations (18)-(20): the summation over s from 1 to infinity does not agree with the state-space {1,2,...,N} considered in equation (17). Since it deals with the finite ring I suggest to use a specific state-space in such equations.
- Page 5, l. 138: provide a reference of the statement that the complexity equals one for a diffusive process
- Page 10, l. 287, write P^0 instead of P^O
- Page 12, l. 378: instead of
M.O. Céceres; Nizama, M.; Pennini, F. Fisher and Shannon, “Functionals for Hyperbolic Diffusion”
write
M.O. Céceres; M. Nizama and F. Pennini, "Fisher and Shannon, Functionals for Hyperbolic Diffusion”
Author Response
Comments 1: “The keywords are missing: three to ten pertinent keywords specific to the article should be included.”
Response 1: Thank you for your suggestion, but I believe the journal already covers that aspect during the editing process.
Comments 2: “ Page 2, l. 74-75, Please, improve/clarify the two sentences, since they seem to be in disagreement.”
Response 2: Thank you for pointing that out. I have already revised that sentence in the manuscript.
Comments 3: “Page 2, eq. (4): if this is the definition of the double Fourier-Laplace transform of P_s(t) then it should be renamed by using a different symbol, for instance \tilde P instead of P in the left-hand-side, in order to avoid confusion with the expression of the conditional probability. Moreover, there is misprint: the argument of the function is (u) rather than (t). ”
Response 3: Thank you for pointing that out. The typo has been fixed. I have chosen to keep the notation P_k(u) as it is commonly used in the literature.
Comments 4: “Page 2, l. 83: I suggest to display the two initial conditions in the center of the line in order to give them more visibility.”
Response 4: Thank you for the suggestion. I have implemented it by placing the initial conditions in the center of the line as a non-labeled equation.
Comments 5: “Page 3, l. 87: The inverse the Laplace -> The inverse of the Fourier-Laplace transform”
Response 5: Thank you for your observation, but in this case, only I perform the Laplace inverse. I have the solution in Fourier and time. That change does not correspond.
Comments 6: “Page 5, the equations (18) and (19) are not recalled in the paper, so that the numbers can be cancelled.”
Response 6: Thank you for the suggestion, I have made the changes.
Comments 7: “Page 5, equations (18)-(20): the summation over s from 1 to infinity does not agree with the state-space {1,2,...,N} considered in equation (17). Since it deals with the finite ring I suggest to use a specific state-space in such equations. ”
Response 7: Thank you for pointing that out. I have revised it in the manuscript.
Comments 8: “Page 5, l. 138: provide a reference of the statement that the complexity equals one for a diffusive process”
Response 8: Thank you for pointing that out. I have included a reference to clarify the information following Equation (23).
Comments 9: “Page 10, l. 287, write P^0 instead of P^O”
Response 9: Thank you for pointing that out. I have corrected the typo.
Comments 10: “Page 12, l. 378: instead of
M.O. Céceres; Nizama, M.; Pennini, F. Fisher and Shannon, “Functionals for Hyperbolic Diffusion”
write
M.O. Ca’ceres; M. Nizama and F. Pennini, "Fisher and Shannon, Functionals for Hyperbolic Diffusion””
Response 10: Thank you for pointing that out. I have rectified the error in the citation.
Round 2
Reviewer 1 Report
Comments and Suggestions for Authors
The manuscript can be accepted for publication.